# The Intratumor Bacterial and Fungal Microbiome Is Characterized by HPV, Smoking, and Alcohol Consumption in Head and Neck Squamous Cell Carcinoma

**DOI:** 10.3390/ijms232113250

**Published:** 2022-10-31

**Authors:** Jaideep Chakladar, Daniel John, Shruti Magesh, Matthew Uzelac, Wei Tse Li, Kypros Dereschuk, Lauren Apostol, Kevin T. Brumund, Jessica-Wang Rodriguez, Weg M. Ongkeko

**Affiliations:** 1Department of Surgery, Division of Otolaryngology-Head and Neck Surgery, University of California, San Diego, CA 92093, USA; 2Research Service, VA San Diego Healthcare System, San Diego, CA 92161, USA; 3Division of Head and Neck Surgery, Department of Surgery, VA San Diego Healthcare System, San Diego, CA 92161, USA; 4Department of Pathology, UC San Diego School of Medicine, San Diego, CA 92093, USA; 5Pathology Service, VA San Diego Healthcare System, San Diego, CA 92161, USA

**Keywords:** HNSCC, microbiome, etiology

## Abstract

Head and neck squamous cell carcinoma (HNSCC) tumor phenotypes and clinical outcomes are significantly influenced by etiological agents, such as HPV infection, smoking, and alcohol consumption. Accordingly, the intratumor microbiome has been increasingly implicated in cancer progression and metastasis. However, few studies characterize the intratumor microbial landscape of HNSCC with respect to these etiological agents. In this study, we aimed to investigate the bacterial and fungal landscape of HNSCC in association with HPV infection, smoking, and alcohol consumption. RNA-sequencing data were extracted from The Cancer Genome Atlas (TCGA) regarding 449 tissue samples and 44 normal samples. Pathoscope 2.0 was used to extract the microbial reads. Microbe abundance was compared to clinical variables, oncogenic signatures, and immune-associated pathways. Our results demonstrated that a similar number of dysregulated microbes was overabundant in smokers and nonsmokers, while heavy drinkers were characterized by an underabundance of dysregulated microbes. Conversely, the majority of dysregulated microbes were overabundant in HPV+ tumor samples when compared to HPV- tumor samples. Moreover, we observed that many dysregulated microbes were associated with oncogenic and metastatic pathways, suggesting their roles in influencing carcinogenesis. These microbes provide insights regarding potential mechanisms for tumor pathogenesis and progression with respect to the three etiological agents.

## 1. Introduction

HNSCC is the seventh most common cancer in the world, accounting for over 930,000 new cases and 467,000 deaths in 2020 [1]. Head and neck cancer includes cancers that develop in the oral cavity, sinuses, nasal cavity, salivary glands, throat, or larynx [2]. The majority of head and neck cancers originate from squamous cells in the mucosal epithelium, eventually developing into head and neck squamous cell carcinoma (HNSCC) [3]. While new treatments, including proton therapies [4,5], immunotherapies [6], and combined therapies [7], have improved the options available to HNSCC patients, they remain primarily experimental and have not resulted in a definitive improvement in patient prognosis in the last 10 years [8]. The global incidence of HNSCC continues to rise and is expected to increase by 30% by 2030 [3]. As such, more studies are needed to understand the underlying mechanisms that promote HNSCC carcinogenesis and, furthermore, improve the preventive measures, diagnosis, and prognosis of HNSCC.

Recent studies of the human microbiome have revealed its crucial role in the development of a variety of human diseases, including arthritis [9,10], inflammatory bowel disease (IBD) [11,12], diabetes [13,14], and cancer [15]. While most studies of the human microbiome investigate disease species found in the gut, investigations into the intratumoral microbiome reveal that species within the tumor microenvironment may also play a crucial role in the prognosis and diagnosis of cancers through the production of small molecule metabolites [16] and the downstream modulation of tumor immunity [17]. The presence of certain intratumor bacteria also affects the efficacy of chemotherapeutic drugs, altering their chemical structures and, consequently, their anti-cancer effects [18]. In previous studies, we have investigated the role of the intratumor microbiome in thyroid [19], bladder [20], pancreas [21], lung [22], and prostate [23] cancers, demonstrating its implications for prognosis, response to anticancer therapy, influence on subtypes, and contribution to carcinogenesis through the modulation of the immune system, methylation, mutations, and more. In oral squamous cell carcinoma (OSCC), the most common malignancies in head and neck cancers, reagrding *F. nucleatum* and *Porphyromonas gingivalis* within the oral cavity, were found to promote tumor progression in mice [24]. A pancancer study conducted on patients from The Cancer Genome Atlas (TCGA) developed a highly accurate diagnostic signature from intratumor microbial reads [25], demonstrating the importance and diagnostic capabilities of microbes present within the HNSCC tumor microenvironment, amongst other carcinomas. Recently, another study developed an eight-panel microbial signature with the ability to distinguish tumor samples from normal tissues using an AUC > 0.7 [26]. Furthermore, these microbes demonstrated significant correlations with clinicopathological variables, including tumor staging, histologic neoplasm grade, and host age.

A defining feature of HNSCC is the distinct phenotypes and varying efficacies of those treatments influenced by etiology. Infection with the human papilloma virus (HPV), tobacco usage, and alcohol consumption are the most influential risk factors of HNSCC [3]. The incidence of HPV in studies of HNSCC patients ranges from 50–90% [27,28,29,30], with HPV-negative HNSCCs experiencing poorer antitumor immune response and prognosis [31]. A total of 70% of new HNSCC diagnoses are associated with either alcohol or tobacco use, with alcohol users displaying significantly poorer overall survival rates [32], and smokers experiencing a 50% reduction in survival time compared to nonsmokers in HNSCC [33]. Synergistic effects between these risk factors have been proposed. For example, studies have found that smoking and HPV infection may interact to promote cancer development and progression [34], with smoking contributing to the carcinogenic effects of HPV. Despite these studies, which explore the distinct oncogenic mechanisms of HPV, tobacco, and alcohol-induced HNSCC, few studies have associated the intratumor microbiome with these risk factors, and none, to our knowledge, have comprehensively classified the microbial landscape in all three etiologies.

In this study, we characterized the similarities and differences in the HNSCC intratumor microbiome according to HPV, smoking, and alcohol etiologies using RNA-sequencing data from TCGA. We first identified those microbes differentially abundant according to these cohorts, evaluated their associations with clinical patient variables, and finally studied their correlations with cancer and immunological pathways (Figure 1). With this study, we aim to characterize the intratumor microbiome in HNSCC across its three defining etiologies and evaluate its clinical significance for patient prognosis and overall influence on carcinogenesis.

## 2. Results

### 2.1. Data Acquisition and Identification of Intratumor Microbes

In order to identify intratumor microbes in head and neck squamous cell carcinoma (HNSCC), we extracted raw RNA-sequencing data from The Cancer Genome Atlas (TCGA). Specifically, we extracted bacterial read counts for 449 primary tissue samples and 44 normal samples, and fungal read counts for 443 primary tissue samples and 43 normal samples (Figure 1). The clinical patient data were obtained from the Broad Institute GDAC Firehose Database (http://gdac.broadinstitute.org/), accessed on 27 September 2022. For the purposes of this project, we investigated three patient etiologies associated with HNSCC occurrence and progression: smoking, alcohol consumption, and HPV infection. Previous studies have suggested that each of these etiologies may independently alter the intratumor microbial landscape of HNSCC [35,36,37], and, as such, we analyzed patient data in four primary cohorts: (1) 449 HNSCC samples and 44 adjacent normals, (2) 99 HPV+ cancer samples and 427 HPV- cancer samples, (3) 153 HPV- smoking cancer samples and 92 HPV- nonsmoking cancer samples, (4) 60 nondrinkers, 97 moderate drinkers, and 63 heavy drinkers, all of which were cancer samples (Figure 1).

### 2.2. Evaluation of Contamination Correction

After successfully extracting bacterial and fungal reads from 493 and 485 patients, respectively, we removed any potential contaminants with three methods of contamination correction. The bacterial and fungal contaminants are shown in Figure 2. First, we identified 22 bacteria and 61 fungal contaminants using correction by sequencing date. Next, we corrected for 22 bacteria and three fungal species that were likely contributing to contamination via the sequencing plate after tissue extraction. Finally, we compiled a list of common hospital and sequencing contaminants from the literature [38,39], of which 175 bacteria species and two fungal species were present and corrected for in our samples (Figure 2). In total, we identified 219 bacteria and 66 fungal contaminants, which were not included in downstream analyses.

Figure 1 shows a schematic of the project workflow. The RNA-sequencing data were downloaded from The Cancer Genome Atlas (TCGA) for 449 HNSCC primary tumor samples and 44 adjacent solid tissue normal samples. The PathoScope 2.0 framework was used to extract Figure 2 (the identification of potential contaminants introduced through the sequencing process). Potential microbial contaminants were identified and removed from further analysis using three contamination correction methods: irregular microbe counts present on specific sequencing plates, irregular microbe counts on specific dates, and commonly known hospital and sequencing contaminants from the literature.

### 2.3. Differential Abundance and Clinical Significance of Intratumor Microbes Associated with HNSCC

Due to the known impact of localization, age, and sex on HNSCC prognosis and outcomes, we first checked for any distinct differences in the microbial landscape of HNSCC according to these factors using principal component analysis (PCA) (Appendix A). We did not observe major differences in microbial presence according to localization, age, or sex.

We performed differential abundance analysis between the tumor and adjacent normal tissue to identify dysregulated microbes associated with HNSCC. Significantly differentially abundant microbes were defined by a *p*-value of <0.05 and a log-fold change (FC) > 1. The Bonferroni method was used to correct the p-values for multiple hypothesis testing. We identified 161 total microbes dysregulated between tumor and normal tissue (Figure 3). The dysregulated microbes (between tumor and normal tissue) were predominantly fungi, which consisted of 143 fungal species and only 18 bacteria species (Figure 3). We found nine fungal species and four bacterial species that were significantly overabundant in tumor tissue. Some of these species include, but are not limited to, *Candidatus Mycoplasma haemolamae*, *Bradyrhizobium japonicum*, *Escherichia coli str. K-12 substr. MG1655*, and *Teratosphaeria gauchensis.* We also identified 134 fungal species and 14 bacterial species overabundant in the normal samples.

Next, we analyzed the abundance of significantly dysregulated microbes among the clinical variables. Comparisons between the microbes and the clinical variables were conducted for all cohorts, with a statistical significance defined as *p*-value < 0.05. Specifically, we examined the comparisons according to the following clinical variables and features: patient age at diagnosis, neoplasm cancer status, perineural invasion, follow-up vital status, pathological stage, and cancer TNM staging.

We found that eight fungal species underabundant in the tumor samples were significantly dysregulated in clinical variable comparisons (Figure 3). Specifically, a decreased abundance of *Aspergillus flavus*, *Coccidioides immitis RS*, and *Gaeumannomyces tritici R3-111a-1* was significantly associated with a lack of perineural invasion, while an increased presence of *Blumeria graminis f.* sp. *Hordei* was significantly associated with early cancer pathological stages. Interestingly, the increased abundance of *Saccharomyces cerevisiae EC1118* and *Podospora anserina S mat+* was correlated with advanced pathological N-stage (NIII) and tumor neoplasm presence, which suggests that the dysregulation of these particular microbes may contribute to cancer pathogenesis and metastasis.

### 2.4. Differential Abundance and Clinical Significance of Intratumor Microbes Associated with HPV Status in HNSCC

We found 31 fungi and two bacteria to be differentially abundant between the HPV+ tumor samples and the HPV- tumor samples (Figure 4). Specifically, our results demonstrated that 26 fungi were overabundant in HPV+ tumor samples when compared to the HPV- tumor samples, suggesting that these microbes may play a role in HPV(-)-specific cancer occurrence. In particular, *Inosperma fulvum*, *Phlyctochytrium arcticum*, and *uncultured Cryptomycota* displayed the strongest statistical significance in differential abundance analyses and comparisons with clinical patient data (Figure 4). In the HPV+ tumor samples, *Inosperma fulvum* abundance is associated with higher pathological T-stage, while *Phlyctochytrium arcticum* abundance is associated with neoplasm presence, and an abundance of *uncultured Cryptomycota* is associated with neoplasm presence and higher pathological staging. Additionally, we also observed that five fungal species and two bacterial species were overabundant in the HPV- tumors samples compared to the HPV+ tumor samples. Of these microbes, an increased abundance of *uncultured bacterium, Myroides orodatus,* and *uncultured Uromyces* is associated with higher pathological staging, while the increased presence of *Morchella esculenta* is associated with patient survival (in vital status follow-up) and an absence of perineural invasion in the HPV- tumor samples. Interestingly, the presence of *Pseudogymnoascus destructans 20631-21* is associated with a lower pathological stage but patient death in vital status follow-up. In total, we found that the abundance of 13 microbes was significantly associated with clinical patient variables. As such, our results show that HPV+ HNSCC and HPV- HNSCC are characterized by distinct intratumor microbial landscapes, which are clinically relevant to patient prognosis.

### 2.5. Differential Abundance and Clinical Significance of Intratumor Microbes Associated with HNSCC Smokers

In our analysis of the HPV- smoking and HPV- nonsmoking tumor samples, we found 10 fungi and eight bacteria to be significantly dysregulated according to smoking status. Of these significantly dysregulated microbes, six bacterial species and seven fungal species were overabundant in the nonsmoking HPV- tumor samples, while two bacterial species and three fungal species were overabundant in the smoking HPV- tumor samples (Figure 5). In particular, *Leptotrichia buccalis C-1013-b* displayed the most significant overabundance in the nonsmoking HPV- tumor samples (*p*-value < 0.001), while *Saccharomyces cervisiae YJM244* displayed the most significant overabundance in the smoking HPV- tumor samples (*p*-value < 0.001). Four microbes, *Leptotrichia buccalis C-1013-b*, *Haemophilus parainfluenzae T3T1*, *Punctularia strigosozonata HHB-11173 SS5*, *and Fomitiporia mediterranea MF3/22,* were overabundant in the HPV- HNSCC nonsmokers and were significantly correlated with clinical patient variables, whereas no overabundant microbes in the HPV+ HNSCC smokers displayed significant correlations with clinical patient variables (Figure 5). Interestingly, the presence of all four microbes was associated with a lower pathological stage in the nonsmoking HPV- tumor samples.

### 2.6. Smoking and HPV(-)-Associated HNSCC Display Distinct Intratumor Microbial Profiles

Previous studies have implicated smoking as a risk factor for worse HNSCC clinical outcomes and patient survival rates [40]. Similarly, data have shown that HPV+ patients face a higher risk of HNSCC occurrence [41]. As such, we aimed to comprehensively analyze commonly dysregulated microbes across both of these respective etiologies when compared to solid tissue normals. Our results demonstrated that two microbes, the bacteria *Candidatus Mycoplasma haemolamae* and the fungus *Saccharomyces cerevisiae YJM244,* were common in both HPV+ HNSCC nonsmokers and HPV- HNSCC smokers (Figure 6). However, the majority of fungal and bacterial microbes, including 27 bacteria and 75 fungi, were uniquely dysregulated in HPV+ patients when compared to HPV- smoking patients, indicating that these two etiologies may uniquely alter the intratumor microbiome of HNSCC during oncogenesis. Further studies are required to fully elucidate the mechanisms by which these microbes may influence clinical outcomes and alter the course of disease progression.

### 2.7. Differential Abundance of Microbes Associated with Alcohol Consumption

Given the established risk of HNSCC and the significant alteration of the oral microbiome in alcohol users [42,43], we compared microbial abundance in nondrinkers, moderate drinkers, and heavy drinkers through differential abundance analysis. Patients were classified as nondrinkers, moderate drinkers, and heavy drinkers by frequency of alcohol consumption. Our results showed that a total of 14 fungi and six bacteria were dysregulated between nondrinkers, moderate drinkers, and heavy drinkers (Figure 7). In particular, we observed an overabundance of *Meiothermus silvanus DSM 9946*, *Acinetobacter baumannii ACICU*, *Ralstonia pickettii 12J*, *Cupriavidus necator H16*, and *Acidovorax* sp. *JS42* in heavy drinkers, while the same microbes were found to be underabundant in moderate drinkers. We additionally identified 14 microbes that were all found to be overabundant in nondrinkers. Comparisons of these microbes with clinical variables yielded no significant results.

### 2.8. Microbe Abundance Correlation to Oncogenic and Tumor-Suppressive Pathways

We used gene set enrichment analysis (GSEA) to correlate microbial abundance with known oncogenic and immunological signatures [44,45]. From the collections available on the Molecular Signatures Database (https://www.gsea-msigdb.org/gsea/msigdb/index.jsp), accessed on 2 September 2022. we classified all available pathways into the following collections from the literature: Antitumor, Cell Cycle/Growth, DNA damage, Inflammation, Metabolism, Metastasis, Oncogenic, Protumor, and Tumor-Suppressive. Only statistically, significantly (*p* < 0.05 and false discovery rate (FDR) < 0.25)-enriched signatures were included.

In both the HNSCC tumor samples and the adjacent normals, the dysregulated fungal microbes disproportionately accounted for the enriched pathways when compared to the dysregulated bacteria species (Figure 8A). The dysregulated fungal microbes were primarily and significantly correlated with the enrichment of cell cycle/growth signatures (*n* = 31,919), followed by those signatures associated with metastasis (*n* = 35,035) in both HNSCC and the adjacent normals. Across HNSCC cancer patients, we observed that many dysregulated fungal microbes and several dysregulated bacterial microbes were associated with the enriched oncogenic, tumor-suppressive, and immune-associated (IA) pathways. Similar trends were observed in the adjacent normal samples as well. As such, we more closely examined the correlations between the dysregulated microbes and the known oncogenic and immune-associated pathways with respect to each of the three main etiologies discussed in this study.

We found that a disproportionately small number of microbes significantly enriched the oncogenic and IA signatures in HPV+ and HPV- HNSCC when compared to the other etiology groups (Figure 8B). In HPV+ HNSCC, 45 bacteria were correlated with enriched signatures pertaining to metabolism. Studies have established that pathways pertaining to metabolism are closely correlated with the production of free radicals, which may subsequently contribute to oncogenic mutations [46]. In addition, oncogene activation may regulate metabolic activity to increase cancer cell growth and survival. As such, we expect that the enriched metabolic pathways unique to HPV+ tumor samples may be regulating other pathways associated with cancer progression or pathogenesis.

*Uncultured bacterium* enriched the oncogenic signatures in both the HPV- HNSCC and HPV+ HNSCC groups. In the HPV- tumor samples, we found that the overabundance of the *uncultured bacterium* was correlated with the upregulation of the REACTOME_OLFACTORY_SIGNALING_PATHWAY oncogenic pathway (Figure 9A), which has been found to contribute to tumorigenesis and the development of cancers [47], as well as to the increased survival and proliferation of cells through the modulation of MAPK, Rho, and AKT signaling cascades [48,49]. We observed that the *uncultured bacterium* was also associated with the upregulation of pathways pertaining to metabolism and carcinogenesis in the HPV+ tumor samples. Therefore, we predict that this particular microbe may play significant roles in cancer pathogenesis and proliferation.

Among the HPV- smoking and nonsmoking HNSCC samples, we observed that the dysregulated microbes were primarily associated with the upregulation of signatures relating to cell cycle growth (smoking: *n* = 451, nonsmoking: *n* = 749), metastasis (smoking = 276, nonsmoking: *n* = 577), and carcinogenesis (smoking: *n* = 279, nonsmoking: *n* = 617) (Figure 8C). Overall, the number of significant microbe-enriched oncogenic and immune-associated (IA) signatures was similar across all nine groups; however, many more bacterial species contributed to the enrichment of these pathways in nonsmokers (*n* = 606) vs. smokers (*n* = 0). In particular, the overabundance of *Agaricus bisporus* in the HPV- smoking tumor samples was correlated with an increased expression of the following pathways: REACTOME_SHC1_EVENTS_IN_ERBB2_SIGNALING, PID_HIF1_PATHWAY, and PID_EPO_PATHWAY (Figure 9B).

These pathways are closely implicated in the maintenance of cancer cells and tumor progression through the expression of transcription factors and the regulation of cellular signaling [50,51]. Similarly, in the HPV- nonsmoking tumor samples, the overabundance of *Fomitiporia mediterranea MF3/22* and *Haemophilus parainfluenzae T3T1* was correlated with the upregulation of several pathways corresponding to carcinogenesis and metastasis (Figure 9B). However, these microbes also correlate to pathways concerning tumor suppression, thus demonstrating the potential of antitumor properties in these microbes.

Our analysis of dysregulated microbes unique to HNSCC heavy drinkers and HNSCC nondrinkers suggested that the oncogenic and IA signatures were enriched by distinct microbial features within each cohort. In HNSCC heavy drinkers, all the significantly enriched pathways were enriched by fungal species (*n* = 102), while bacteria species predominantly contributed to the enriched pathways in nondrinkers (bacteria: *n* = 2578, fungi: *n* = 113) (Figure 8D). Interestingly, antitumor pathways were primarily enriched in the HNSCC nondrinkers (bacteria: *n* = 905, fungi: *n* = 43), followed by pathways associated with metabolism (bacteria: *n* = 674, fungi: *n* = 0). In HNSCC heavy drinkers, the oncogenic (*n* = 35) and antitumor pathways (*n* = 25) were primarily enriched by fungal species, although a higher prevalence of enriched oncogenic (bacteria: *n* = 537, fungi = 18) and antitumor (bacteria: *n* = 905, fungi = 43) pathways was observed in the HNSCC nondrinkers. Notably, we observed that the abundance of *Gromochytrium mamkaevae* in heavy drinkers was closely associated with the REACTOME_MET_PROMOTES_CELL_MOTILITY and REACTOME_INTEGRIN_CELL_SURFACE_INTERACTIONS oncogenic signatures (Figure 9C). Previous studies have widely established that these oncogenic signatures play critical roles in promoting the invasiveness and metastasis of cancer cells, primarily through MET and integrin interactions [52,53].

### 2.9. Classification of Microbes as Predictors of Increased and Decreased Carcinogenesis

We further classified the GSEA results to identify those microbes associated with increased and decreased carcinogenesis (Figure 10). We defined microbes associated with decreased carcinogenesis as those that upregulate tumor-suppressive pathways, downregulate oncogenic pathways, and show better patient prognosis (as determined by correlations to increased patient survival outcomes), while those microbes associated with increased carcinogenesis were defined as those which downregulate tumor-suppressive pathways, upregulate oncogenic pathways, and show poor patient prognosis (as determined by correlations to worse patient survival outcomes). All findings for the oncogenic pathways and tumor-suppressive pathways are from GSEA. In the HPV- HNSCC smokers, 15 bacteria and 28 fungi in total were associated with decreased carcinogenesis, while 11 bacteria and 15 fungi were associated with increased carcinogenesis. Specifically, we found that 11 bacteria and four fungi were uniquely associated with the downregulation of several oncogenic pathways, including pathways associated with metastasis pathways, inflammation, and DNA damage. In addition, our results indicated that 22 fungi and two bacteria were correlated with better patient prognosis overall. We found that the bacterial microbe *Haemophilus parainfluenzae T3T1* is correlated with all three mechanisms of decreased carcinogenesis, while *Leptotrichia buccalis C-1013-b* is associated with downregulated oncogenic pathways and better patient prognosis, and *Fomitiporia mediterranea MF3/22* is associated with upregulated tumor-suppressive pathways and better patient prognosis. Additionally, we found 11 bacteria and 15 fungal microbes associated with increased carcinogenesis. A total of four unique fungal species were associated with the upregulation of oncogenic pathways pertaining to metastasis, cell growth, and inflammation. Moreover, we observed that four fungal species and six bacterial species were correlated with the downregulation of tumor-suppressive pathways, while 10 unique microbes were identified to be representative of a worse overall prognosis.

Notably, we found that *Punctularia strigosozonata HHB-11173 SS5* is associated with upregulated oncogenic pathways and worsened patient prognosis. Interestingly, the microbe *Haemophilus parainfluenzae T3T1* was also associated with decreased carcinogenesis, specifically, the upregulation of oncogenic pathways and downregulation of tumor-suppressive pathways. Given the dual role of *Haemophilus parainfluenzae T3T1* as both a cancer-promoting and -inhibiting microbe, further studies are required to fully elucidate the mechanisms and circumstances in which it may act as a cancer promoter or inhibitor.

## 3. Discussion

In this study, we characterized the intratumor microbiome in HNSCC according to its most significant etiologies: HPV infection, tobacco smoking, and alcohol consumption. Previous studies have demonstrated that tumor profiles, mutations, and infiltrating immune cells are uniquely modulated by HPV status [54,55] and tobacco smoking [37,56]; these etiologies have also been compared to differing clinical patient outcomes, depending on infection status and severity of smoking behavioral habits. Additionally, studies have demonstrated that HNSCC patients with a history of heavy alcohol consumption express select microRNAs (miR) within the tumor microenvironment, which are subsequently implicated with poor patient prognosis. Specifically, miR-375, miR-21, miR-30a, and miR-934 are associated with poor patient survival and tumor occurrence [57,58]. In addition, patients with a history of heavy alcohol consumption are associated with worse prognosis and response to immunotherapies [35], when compared to those with little to moderate drinking history.

Microbes residing within the oral cavity and the nearby tissues are also significantly dysregulated according to these etiologies. In a study of 495 patients, those with oral HPV exhibited significant oral microbiome beta-diversity when compared to those without HPV, specifically, with a higher abundance of *Actinomycetaceae, Prevotellaceae, Veillonellaceae, Campylobacteraceae,* and *Bacteroidetes* and a lower abundance of *Gemellaceae* [59]. Furthermore, in a study of 1204 adults, overall oral microbiome composition significantly differed between current smokers and people who had never smoked, including a depletion of the *Proteobacteria* phylum in smokers, among other dysregulated species [60]. Studies have also found that the alpha- and beta-diversity of the oral microbiome also differs significantly between heavy drinkers and nondrinkers [43]. However, despite the known dysregulation of the oral microbial environment in association with HPV infection, smoking, and alcohol use, few studies have analyzed the intratumor microbiome in HNSCC with regard to these etiological agents. As such, we aimed to characterize the presence of the intratumor microbiome in HNSCC according to HPV infection, smoking, and alcohol consumption. We also investigated the comparisons between dysregulated microbes in each cohort and clinical patient variables in order to better understand the implications of the microbes in relation to cancer pathogenesis and clinical outcomes. Finally, we aimed to elucidate the specific oncogenic and immunological signatures associated with the relevant microbes in each cohort.

In this study, we extracted RNA-sequencing from TCGA for 449 HNSCC primary tumor samples and 44 adjacent solid tissue normal samples. After removing any potential contaminants, we found that the vast majority of the dysregulated microbes were overabundant in the normal samples when compared to the tumor samples. Our results are consistent with previous findings that microbial species are underabundant in tumor tissue when compared to adjacent normal tissue. We additionally found that several fungal microbes were relevant to pathological staging. In particular, *Saccharomyces Cerevisiae YJM195* was positively associated with advanced pathological N-stage. However, evidence has shown that *Saccharomyces Cerevisiae* species have a tumor-suppressive role in colorectal cancer and are implicated in cell growth inhibition, B cell activation, and epithelial cell apoptosis [61,62]. Thus, further studies are required to elucidate the complex interactions and metabolism of *Saccharomyces Cerevisiae* within the HNSCC tumor microenvironment, which was negatively associated with patient prognosis in our study.

While current studies on the microbiome and HPV infection in HNSCC are limited, one study reported the overabundance of several bacterial species in HPV+ HNSCC saliva samples [63]. As such, we investigated microbial abundance in the HPV+ tumor and HPV- tumor samples, as well as their associations with clinical patient variables. Although patients with HPV- HNSCC tend to experience poorer prognosis [31], our results indicate that bacterial and fungal species may have more prognostic relevance in HPV+ HNSCC, due to an increased dysregulation and association with clinical variables such as pathological staging, neoplasm presence, and vital status. Fungal species, in particular, contributed to the majority of the dysregulated species between HPV+ and HPV- HNSCC.

Interestingly, we found that the majority of dysregulated fungal and bacterial microbes were overabundant in HPV+ tumor tissue when compared to HPV- tumor tissue. Specifically, we found that *uncultured Cryptomycota*, an overabundant fungus in HPV+ HNSCC samples, was associated with neoplasm presence and a higher pathological stage. As a newly discovered phylum, many *Cryptomycota* species have yet to be cultured. We predict that this novel microbe may have critical implications in the modulation of cancer outcomes and may act as a biomarker for HPV-associated HNSCC. However, further studies are required to fully understand the role that this microbe may play in cancer onset and progression.

When considering HPV- smoking and HPV- nonsmoking comparisons, we observed that the direction of microbial dysregulation was more balanced, with a similar number of microbes overabundant in both the smokers and nonsmokers. Only those microbes significantly overabundant in nonsmokers, however, were associated with clinical variables. Interestingly, all clinically significant microbes were associated with a lower pathological stage, indicating that these microbes may play protective roles in patient prognosis. Given that smoking is known to deplete and alter the function of microbes in the oral epithelia [60], it is plausible that certain interactions within the tumor microenvironment relevant to patient prognosis may be inhibited in smokers. Furthermore, when we compared the dysregulated microbes in smoking and HPV+ patients, only two species, *Candidatus Mycoplasms haemolamae* and *Saccharomyces cerevisiae YJM244*, were commonly dysregulated, whereas the majority of the dysregulated microbes were unique to smoking or HPV+ patients. Further studies invitro are needed to validate the mechanisms that contribute to the distinct microbial landscapes observed in these patients.

Furthermore, our findings suggest that HNSCC-heavy drinkers experience an underabundance of bacterial and fungal species. Despite previous studies that have noted significant differences in the oral microbiome between drinkers and nondrinkers [64], our findings suggest a more detailed understanding: the microbial abundance in the tumor microenvironment is similar between heavy and moderate drinkers, but significantly different when compared to nondrinkers. Interactions between microbial species in moderate and heavy alcoholic conditions must be further studied to understand their contributions to tumor pathology and oncogenesis.

While most studies of the oral microbiome and intratumor microbiome in HNSCC focus on those bacterial species sequenced through 16s sequencing, we found that microbial dysregulation in cancer, HPV, and smoking were characterized primarily by a dysregulation in the fungal species. Fungal species additionally accounted for the vast majority of the enriched oncogenic and immune-associated signatures across these etiological cohorts. However, among the alcohol cohorts, we observed that, primarily, the fungal species contributed to the enriched oncogenic and immune-associated pathways in heavy drinkers, while the bacterial species contributed to the majority of the enriched pathways in nondrinkers. As such, further studies should investigate the mechanisms by which alcohol consumption may modulate the functional pathways and interactions of bacteria and fungi in the intratumor microbiome, with an emphasis on the oncogenic effects of fungal species in drinkers and bacterial species in nondrinkers.

Our analysis of the oncogenic and immunological signatures revealed that *Haemophilus parainfluenzae T3T1* is associated with signatures that both regulate increased and decreased carcinogenic activity. *Haemophilus parainfluenzae* is a normal oral florum, a known inflammatory agent, and the cause of several acute diseases in the respiratory tract [65]. Additionally, intratumor *Haemophilus parainfluenzae* is associated with an increased risk of EGFR+ nonsmall cell lung cancer [66] and may possibly drive lung cancer oncogenesis through NTHi-driven inflammation [67]. While our findings suggest that this microbe may play a crucial role in procancer and anticancer pathways, further invitro experiments are needed to reveal which mechanisms modulate its role in oncogenesis in HNSCC.

In all, our findings suggest that the intratumor microbiome in HNSCC differs greatly according to HPV infection, smoking history, and alcohol consumption. Previous studies of the intratumor microbiome in HNSCC have largely focused on bacterial species through 16s sequencing. To the best of our knowledge, we are the first to investigate the fungal microbial landscape across these three primary etiologies. Subsequently, the fungal microbial landscape appeared to have a more characteristic role in etiology-based HNSCC. In particular, we identified overabundant microbes in each etiology, some of which were associated with clinical patient and prognostic variables. Furthermore, microbial species demonstrated unique correlations between oncogenic and immunological pathways across these cohorts, indicating their unique involvement and roles in cancer pathogenesis within the tumor microenvironment according to these etiologies. Nonetheless, the results in our study are correlative; therefore, further invitro validation is necessary to elucidate the exact role and mechanisms by which these microbes may contribute to tumor growth or inhibition according to these etiologies.

## 4. Materials and Methods

### 4.1. Data Acquisition

Raw whole transcriptome RNA-sequencing data were downloaded from The Cancer Genome Atlas (TCGA) for 528 HNSCC primary tumor samples and 44 adjacent solid tissue normal samples. Clinical patient data were obtained from the Broad Genome Data Analysis Center (GDAC) Firehose.

### 4.2. Extraction of Bacterial and Fungal Read Counts

RNA-sequencing data were filtered for bacterial and fungal reads through direct alignment to a reference library of known bacterial and fungal signatures, using the PathoScope 2.0 framework and the NCBI nucleotide database. A total of 79 tumor samples failed to yield successful alignments for bacterial genomes, and 85 tumor samples and one normal sample failed to yield successful alignments for fungal genomes and were thus eliminated from the dataset. Our final dataset consisted of 449 primary tissue samples and 44 normal samples for analyses concerning bacteria, and 443 primary tissue samples and 43 normal samples for analyses concerning fungi. Aitchonson’s log-ratio transformation was conducted to normalize extracted read counts and reduce variability between samples.

We analyzed all data in the following four cohorts based on etiology to limit the effect of confounding variables: (1) 449 HNSCC samples and 44 adjacent normals, (2) 99 HPV+ cancer samples and 427 HPV- cancer samples, (3) 153 HPV- smoking cancer samples and 92 HPV- nonsmoking cancer samples, and (4) 60 nondrinkers, 97 moderate drinkers, and 63 heavy drinkers, all of which were cancer samples. For the HPV cohorts, only HPV+ or HPV- patients were included. Some of our patients did not have either an HPV+ or HPV- diagnosis so they were excluded. All clinical information was extracted from the clinical patient file, which was obtained from the Broad Genome Data Analysis Center (GDAC) Firehose. We did not analyze HPV- nonsmoking samples for those analyses concerning nondrinkers, moderate drinkers, and heavy drinkers, as we did not have a substantial number of patients to do so. We categorized nondrinkers, moderate drinkers, and heavy drinkers according to the frequency of alcohol consumption, as defined by Urashima et al. [68]. For the purposes of this study, we classified nondrinkers as zero drinks a day, moderate drinkers as one drink a day, and heavy drinkers as two or more drinks a day.

### 4.3. Evaluation of Contamination

In order to account for contamination in our samples, we conducted contamination correction by sequencing date and plate to identify potential contaminants and remove them from our datasets. First, boxplots were created to correlate microbial abundance to each respective sequencing plate. Microbes that were disproportionately overabundant across sequencing plates were identified as potential contaminants. Next, scatterplots were created to correlate microbial abundance with date of sequencing. Microbes that were disproportionately overabundant for less than three sequencing dates were identified as potential contaminants. Lastly, we compiled a list of common hospital and sequencing contaminants from previously published studies. We identified and removed these respective contaminants if they were found to be present in our datasets.

### 4.4. Differential Abundance between Etiologically Defined Cohorts

Differential abundance analysis was conducted on the following four defined patient cohorts: (1) primary tumor samples and adjacent normal samples, (2) HPV+ cancer samples and HPV- cancer samples, (3) HPV- smoking cancer samples and HPV- nonsmoking cancer samples, and (4) nondrinkers and heavy drinkers, both of which were cancer samples. Differential abundance analysis was conducted on each of the cohorts for bacterial and fungal data, using the Kruskal–Wallis testing in the edge-R library. Statistically significant results were defined as *p*-value < 0.05. Significantly dysregulated microbes were used in further analyses.

We also compared differentially abundant microbes in the following two cohorts: (1) HPV+ cancer samples and HPV- cancer samples and (2) HPV- smoking cancer samples and HPV- nonsmoking cancer samples. In order to do so, we examined our differential abundance analysis results for each of these independent cohorts. We then compared the expression of these differentially abundant microbes in each of the two cohorts. We were able to identify microbes that were uniquely abundant in HPV- HNSCC smokers or HPV+ HNSCC nonsmokers. We were also able to identify microbes that were common to both of these independent cohorts.

### 4.5. Clinical Variable Comparisons to Microbial Abundance in Etiologically Defined Cohorts

We compared significantly dysregulated microbes to clinical variables using the Kruskal–Wallis testing in the edge-R library (*p*-value < 0.05). We also adjusted our p-values with the Bonferroni method to correct for multiple hypothesis testing. For the purposes of this study, we examined comparisons to six main clinical variables and features: patient age at diagnosis, neoplasm cancer status, perineural invasion, follow-up vital status, pathologic stage, and cancer TNM staging. Specifically, we examined the comparisons of dysregulated microbes with clinical variables in the following cohorts: HPV+ tumor samples, HPV- tumor samples, HPV- smokers, HPV- nonsmokers, heavy drinkers, and nondrinkers. We did not observe significant clinical variable comparisons for our drinking cohorts, and as such, we did not include them in our results.

### 4.6. Correlation of Microbial Abundance to Oncogenic and Immunological Signatures

We conducted Gene Set Enrichment Analysis (GSEA) to correlate microbial abundance to known oncogenic and immunological pathways and signatures. Three primary input files were used for GSEA: the expression file, the phenotype file, and the gene set file. The expression file contained the abundance of the microbes in our sequencing data, the phenotype file contained expression of IA genes, and the gene set file contained differentially abundant microbes that could be targeting IA genes. We used the following GSEA parameters: the Pearson metric for ranking genes, one gene set at a time, and the minimum size for exclusion of the smaller sets as 0. GSEA signatures were from the C2, C6, and C7 collections from the Molecular Signatures Database. We classified all signatures and pathways into the following categories based on a literature search: Antitumor, Cell Cycle/Growth, DNA damage, Inflammation, Metabolism, Metastasis, Oncogenic, Protumor, and Tumor-Suppressive. The specific classification files are included in Appendix A. These categories are representative of gene families, or groups of gene sets and pathways which share common features and functions. Only statistically significantly enriched signatures were further analyzed (*p* < 0.05 and false discovery rate (FDR) < 0.25).

## 5. Conclusions

In conclusion, our study significantly advances the understanding of intratumor microbial expression in head and neck squamous cell carcinoma. The intratumor microbiome in HNSCC, across cancer samples, and in HPV, tobacco smoking, and alcohol consumption etiologies were compared with clinical patient variables pertaining to cancer prognosis and occurrence. Overall, the microbial signatures are underabundant in tumor tissue compared to adjacent normals, and the majority of the dysregulated microbes across cancer and etiology comparisons are fungal species. Specifically, *Saccharomyces cerevisiae EC1118* is underabundant in tumor tissue and associated with a higher pathological staging. *Uncultured Cryptomycota* is overabundant in HPV+ nonsmoking tumor samples and is associated with neoplasm presence and a higher pathological stage. When comparing the HPV-, smoking, and nonsmoking samples, we found that *Fomitiporia mediterranea MF3/22* was overabundant in the nonsmoking tumor samples and was associated with a lower pathological stage. We found that *Uncultured Cryptomycota* is also significantly overabundant in nondrinkers. Interestingly, we also found that *Saccharomyces cerevisiae YJM244* is commonly dysregulated in smokers and HPV+ patients when compared to normal tissue, suggesting potential commonalities in their respective microbial landscapes. We found that fungal species primarily enrich the oncogenic and immunological signatures across these etiologies. Within the alcohol comparisons, however, it was found that fungal species primarily enrich these pathways in heavy drinkers, while bacterial species primarily enrich similar pathways in nondrinkers. Finally, *Haemophilus parainfluenzae T3T1* was associated with those enriched signatures that promote cases of both increased and decreased carcinogenic activity. Further invitro experiments are needed to validate these differences within these microbial landscapes and further elucidate the mechanisms underlying microbial activity and oncogenesis.

## Figures and Tables

**Figure 1 ijms-23-13250-f001:**
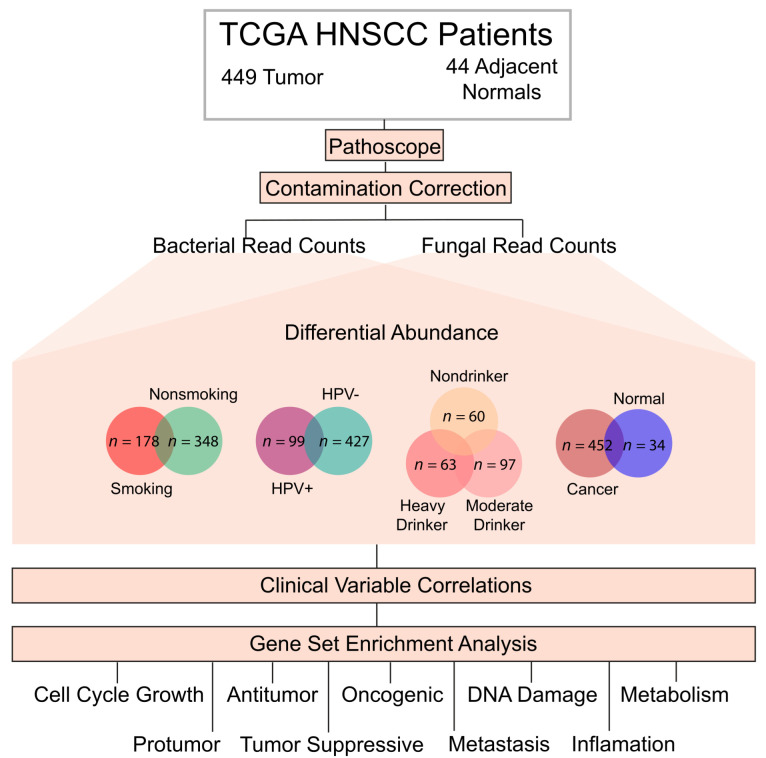
Schematic of project workflow. RNA sequencing data were downloaded from The Cancer Genome Atlas (TCGA) for 449 HNSCC primary tumor samples and 44 adjacent solid tissue normal samples. The PathoScope 2.0 framework was used to extract bacterial and fungal read counts from patient samples. The following four cohorts were used for all data analysis to limit the effect of the confounding variables: (1) 449 HNSCC samples and 44 adjacent normals, (2) 99 HPV+ cancer samples and 427 HPV- cancer samples, (3) 153 HPV- smoking cancer samples and 92 HPV- nonsmoking cancer samples, and (4) 60 nondrinkers, 97 moderate drinkers, and 63 heavy drinkers, all of which were cancer samples. Differential abundance analysis was conducted to identify significantly dysregulated microbes for each of the four cohorts using the Kruskal–Wallis test in the edge-R library (*p*-value < 0.05). Significantly dysregulated microbes were then utilized in clinical variable analyses. Dysregulated microbes were also correlated to oncogenic and immunological signatures through Gene Set Enrichment Analysis (GSEA). Specifically, we classified all relevant signatures and pathways into the following nine categories for analysis: Antitumor, Cell Cycle/Growth, DNA damage, Inflammation, Metabolism, Metastasis, Oncogenic, Protumor, and Tumor-Suppressive.

**Figure 2 ijms-23-13250-f002:**
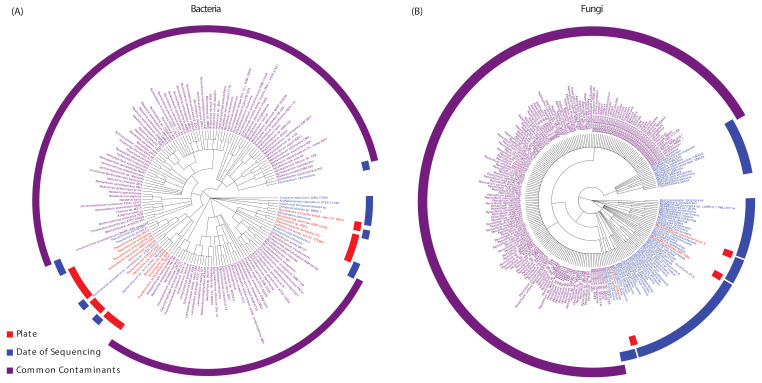
Identification of potential contaminants introduced through the sequencing process. Potential microbial contaminants were identified and removed from further analysis using three contamination correction methods: irregular microbe counts present on specific sequencing plates, irregular microbe counts on specific dates, and commonly known hospital and sequencing contaminants from the literature. (**A**) Potential bacteria contaminants, (**B**) Potential fungal contaminants.

**Figure 3 ijms-23-13250-f003:**
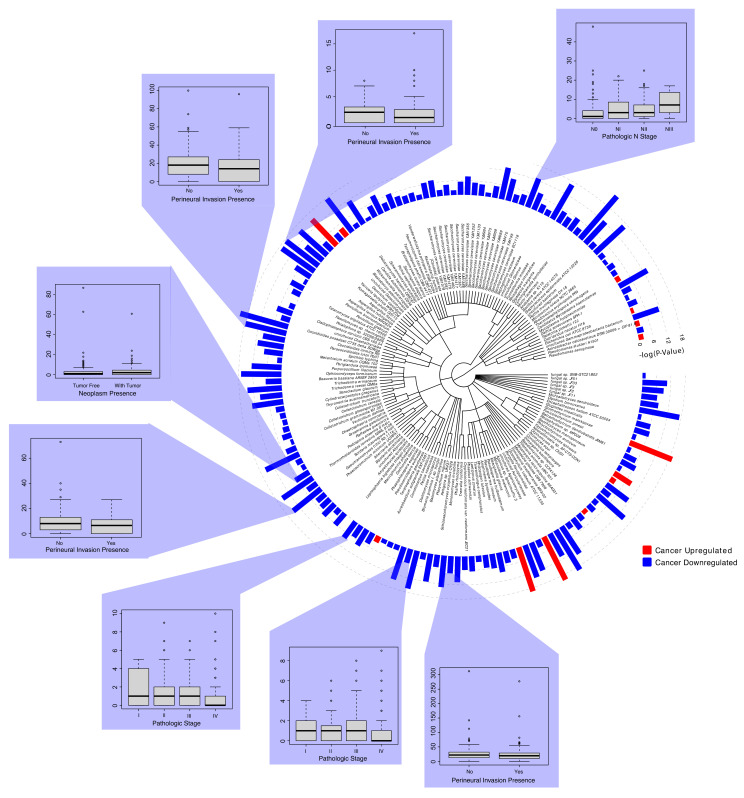
Differential abundance and clinical significance of microbes associated with tumor samples. Significantly dysregulated microbes in HNSCC primary tissue samples are displayed with associations to clinical variables. Differential abundance correlations were conducted to compare microbial abundance between primary tumor samples and adjacent normal samples (*p*-value < 0.05). Red corresponds to upregulation in cancer samples, while blue corresponds to downregulation in the cancer samples. Clinical variable comparisons were conducted for significantly dysregulated microbes using the Kruskal–Wallis tests in the edge-R library (*p*-value < 0.05). The clinical variables investigated included pathological N-stage, perineural invasion presence, neoplasm presence, pathological stage, and vital status.

**Figure 4 ijms-23-13250-f004:**
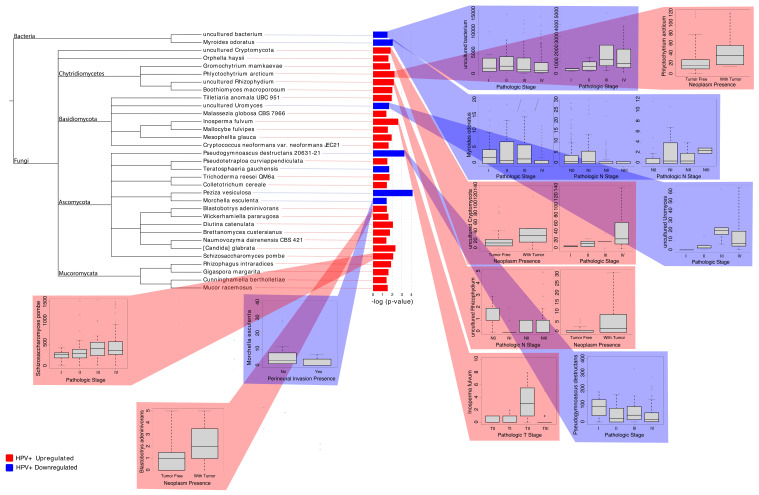
Differential abundance and clinical significance of microbes associated with HPV HNSCC. Significantly dysregulated microbes in HPV+ tumor samples and HPV- tumor samples are displayed with their associations with clinical variables. Red corresponds to overabundance in HPV+ tumor samples, while blue corresponds to overabundance in HPV- tumor samples. The Kruskal–Wallis test in the edge-R library was used to conduct clinical variable associations (*p*-value < 0.05). The specific clinical variables investigated include pathologic stage, vital status, neoplasm presence, perineural invasion presence, pathological T-stage, and pathological N-stage. The various trends in the plots suggest that dysregulated microbes in HPV+ tumor samples are closely associated with neoplasm status and advanced cancer pathologic stage, thus indicating their potential roles in cancer pathogenesis and progression.

**Figure 5 ijms-23-13250-f005:**
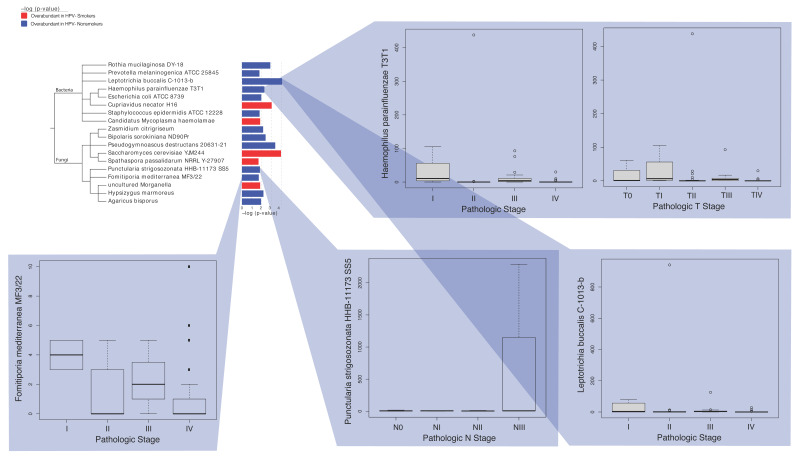
Clinical significance of microbes associated with HPV- smoking and HPV- nonsmoking cohorts. Significantly dysregulated microbes associated with HPV- smoking samples and HPV- nonsmoking samples are displayed with their associations with clinical variables. Red corresponds to overabundance in HPV- smokers, while blue corresponds to overabundance in HPV- nonsmokers. The Kruskal–Wallis test in the edge-R library was used to conduct clinical variable comparisons (*p*-value < 0.05). In particular, we observed significant clinical variable comparisons for the following variables: pathological stage, pathological T-stage, and pathological N-stage. The trends in the plots suggest that the dysregulated microbes in HPV- nonsmokers are associated with a lower pathological stage, thus demonstrating that they may have tumor-suppressive properties.

**Figure 6 ijms-23-13250-f006:**
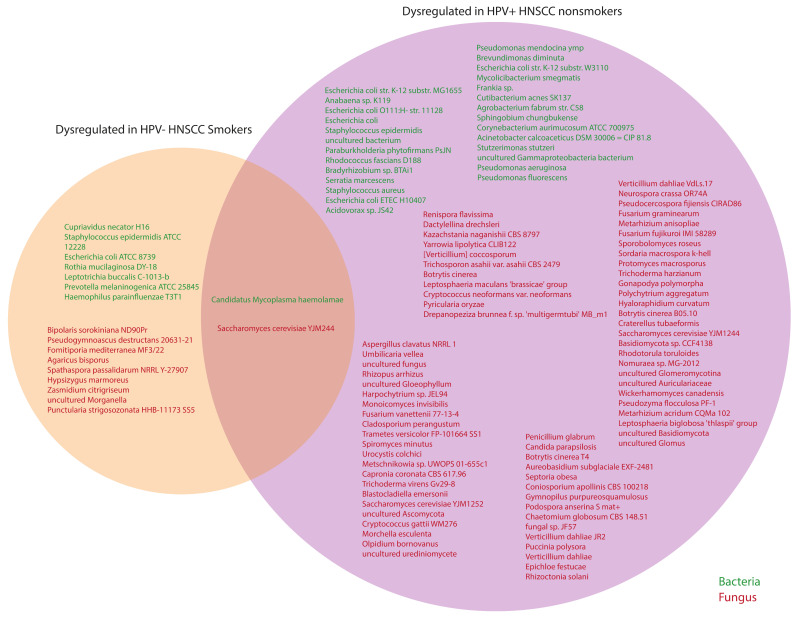
Dysregulation of microbes in HPV- smokers and the HPV+ tumor samples. Venn diagram comparing significantly dysregulated microbes in HPV- HNSCC smokers and HPV+ tumor samples. Green microbes represent bacteria, while red microbes represent fungi. The bacteria *Candidatus Mycoplasma haemolamae* and the fungus *Saccharomyces cerevisiae YJM244* were common to both HPV+ HNSCC nonsmokers and HPV- HNSCC smokers.

**Figure 7 ijms-23-13250-f007:**
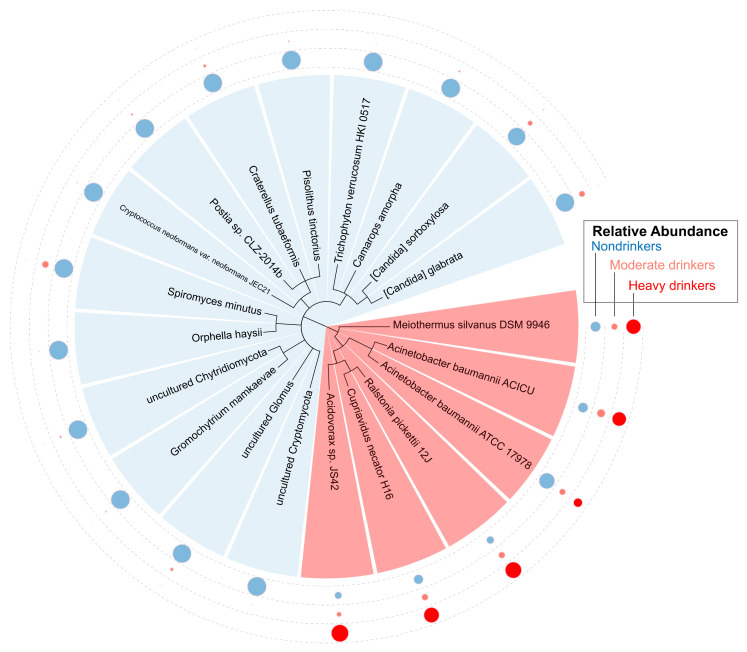
Differential abundance of microbes associated with HPV- heavy drinkers, moderate drinkers, and nondrinkers. Differential abundance analysis was conducted to identify those significantly dysregulated microbes specific to heavy drinkers, moderate drinkers, and nondrinkers. We classified nondrinkers as 0 drinks a day, moderate drinkers as one drink a day, and heavy drinkers as two or more drinks a day. All red species on the phylogenetic tree are classified as bacteria, while the blue species are classified as fungi. Dysregulation of microbial species in association with alcohol consumption habits is represented by the smaller circles surrounding the phylogenetic tree. These smaller circles are organized by cohort: dark red represents heavy drinkers, light red represents moderate drinkers, and light blue represents nondrinkers. If the size of the circle is relatively large, the microbial species is upregulated in that particular cohort. If the size of the circle is relatively small, the microbial species is downregulated in that particular cohort.

**Figure 8 ijms-23-13250-f008:**
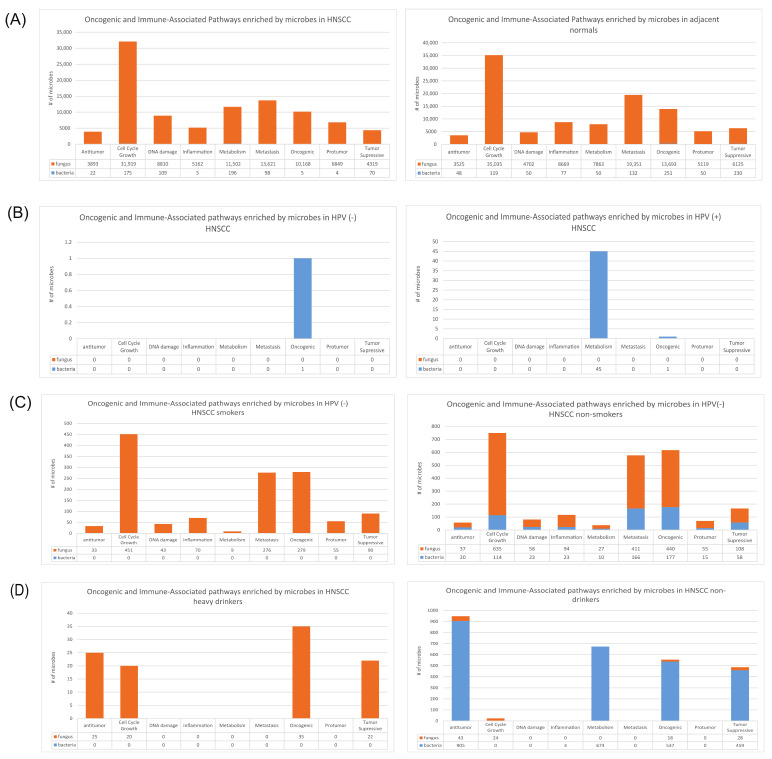
Enriched oncogenic and immune-associated signatures associated with dysregulated microbes. Gene set enrichment analysis (GSEA) was conducted to identify unique oncogenic and immune-associated pathways associated with dysregulated microbes (*p* < 0.05 and false discovery rates (FDRs) < 0.25). GSEA signatures were classified into the following categories for analysis purposes: antitumor, Cell Cycle/Growth, DNA damage, Inflammation, Metabolism, Metastasis, Oncogenic, Protumor, and Tumor-Suppressive. Orange corresponds to fungal microbes, while blue corresponds to bacterial microbes. (**A**) Bar plots visualizing pathway enrichment for dysregulated microbes in HNSCC tumor samples and adjacent normal samples. Dysregulated fungi were found to disproportionately account for enriched pathways when compared to bacteria across both tumor and normal samples. (**B**) Bar plots visualizing pathway enrichment for dysregulated microbes in the HPV- and HPV+ HNSCC samples. Only one bacterial microbe was significantly correlated with pathways for the HPV- HNSCC cohort, while 45 bacteria were correlated with enriched signatures pertaining to metabolism and carcinogenesis in the HPV+ HNSCC cohort. (**C**) Bar plots visualizing pathway enrichment for dysregulated microbes in HPV- HNSCC smokers and HPV- HNSCC nonsmokers. Among the HPV- smoking and nonsmoking HNSCC samples, we observed that dysregulated microbes were primarily associated with the upregulation of signatures relating to cell cycle growth, metastasis, and carcinogenesis. (**D**) Bar plots visualizing pathway enrichment for dysregulated microbes in HNSCC heavy drinkers and HNSCC nondrinkers. In HNSCC heavy drinkers, all significantly enriched pathways were enriched by fungal species, while bacterial species contributed to enriched pathways in nondrinkers.

**Figure 9 ijms-23-13250-f009:**
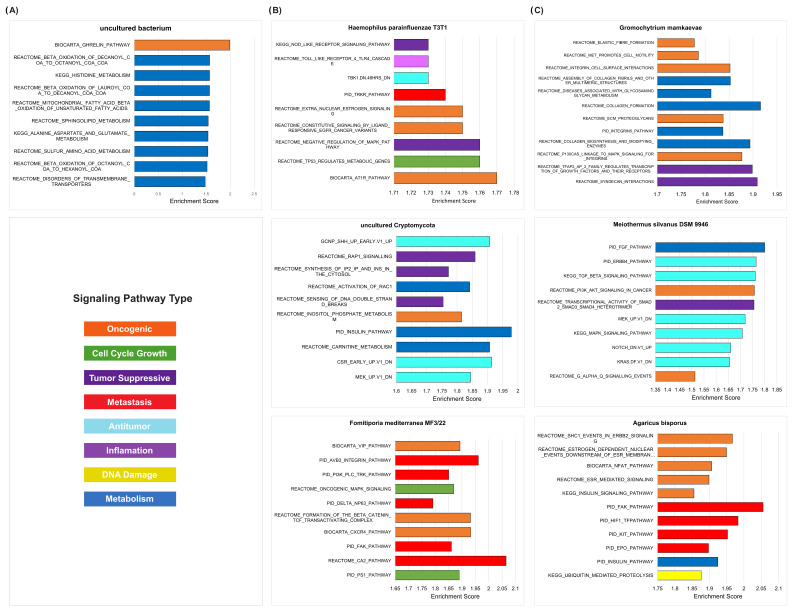
GSEA pathways for microbes from etiological cohorts. Specific GSEA pathways and signatures are provided for by selected microbes from each of the three main etiological cohorts: HPV-infection, smoking, and alcohol consumption. All selected pathways are statistically significant (*p* < 0.05 and false discovery rate (FDR) < 0.25). For all bar graphs, the abundance of the microbes is correlated with the upregulation of the pathways presented. (**A**) Bar plot visualizing enriched pathways for *uncultured bacterium*, which is a dysregulated microbe in the HPV+ tumor samples. (**B**) Bar plots visualizing enriched pathways for microbes in the HPV- smoking and HPV- nonsmoking samples. Specifically, *Haemophilus parainfluenzae T3T1* and *Fomitiporia mediterranea MF3/22* are dysregulated microbes in the HPV- nonsmoking samples, while *Agaricus bisporus* is a dysregulated microbe in the HPV- smoking samples. (**C**) Bar plots visualizing enriched pathways for microbes in the heavy drinker and nondrinker cohorts. *Gromochytrium mamkaevae* is a dysregulated microbe specific to heavy drinkers, while *Meiothermus silvanus DSM 9946* and *uncultured Cryptomycota* are dysregulated microbes specific to nondrinkers.

**Figure 10 ijms-23-13250-f010:**
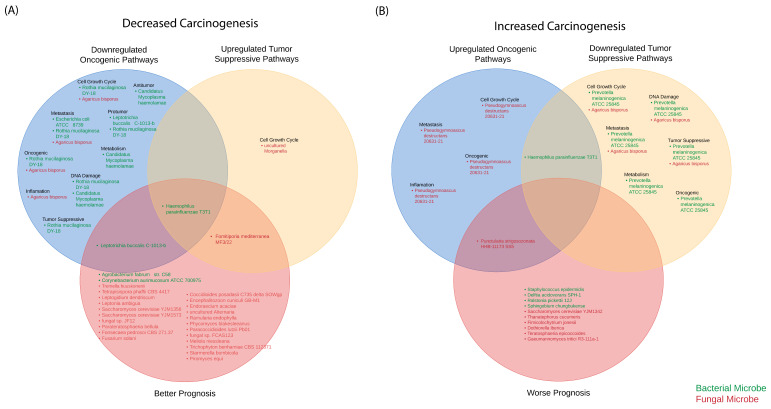
Identification of microbes as predictors of increased or decreased carcinogenesis. Venn diagrams comparing significantly dysregulated microbes in association with pathways relevant to increased or decreased carcinogenesis. Green corresponds to bacterial microbes, while red corresponds to fungal microbes. (**A**) Decreased carcinogenesis was characterized by upregulation of tumor-suppressive pathways, downregulation of oncogenic pathways, and better patient prognosis, as determined by correlations with increased patient survival outcomes. Dysregulated microbes, associated with pathways pertaining to decreased carcinogenesis, are represented in the Venn diagram. *Haemophilus parainfluenzae T3T1* was implicated in all three pathways relevant to decreased carcinogenesis. (**B**) Increased carcinogenesis was characterized by the downregulation of tumor-suppressive pathways, upregulation of oncogenic pathways, and worse patient prognosis, as determined by correlations with decreased patient survival outcomes. Dysregulated microbes associated with those pathways pertaining to increased carcinogenesis are represented in the Venn diagram. *Punctularia strigosozonata HHB-11173 SS5* and *Haemophilus parainfluenzae T3T1* were implicated in multiple pathways concerning increased carcinogenesis and may thus be representative of worse clinical outcomes.

## Data Availability

Publicly available datasets were analyzed in this study. This data can be found here: https://portal.gdc.cancer.gov/projects/TCGA-HNSC.

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
