# Peer review of "The Intratumor Bacterial and Fungal Microbiome Is Characterized by HPV, Smoking, and Alcohol Consumption in Head and Neck Squamous Cell Carcinoma"

_ijms, 2022, doi:10.3390/ijms232113250_

Round 1

Reviewer 1 Report

This study extracted raw RNA-seq data of 528 HNSCC primary tumor samples from The Cancer Genome Atlas, identified the dysregulated microbes and enriched pathways under different cohort comparisons.

Issues:

1) Kruskal-Wallis test is meant for testing differences among groups (not for testing correlation). Thus the authors' claims such as "correlations between significantly dysregulated microbes and clinical variables", "...positively correlated with advanced pathologic N stage" etc are not accurate.

2) The authors described association of microbes with clinical variables when the comparison achieved statistical significance (p-value <0.05). Besides statistical significance, can the authors set a threshold for, or at least discuss the extent of differences (e.g. 10% or 10 fold increases/decreases) in relation to the likelihood of causing any biological influence to the host? And also discuss the detected abundance/differences in relation to noise and detection limit of the platform? For example, in Figure 5, it is shown that Fomitiporia mediterranea MF3/22 in pathological stage I vs pathological stage 5 of HPV- HNSCC nonsmokers were just at median level of 4 vs median level of 0. Is this difference likely to be biological meaningful or could it be just noise?

3) more description is needed about how GSEA was conducted to identify correlation of microbes with gene sets, why count is reported (Figure 8) rather than the enrichment scores (more common metrics) and the text within Figure 9 are incomprehensible.

4) it is unclear what methods were used to classify signatures/pathways and generate findings shown in Figure 10.

Minor issues:

1) The authors stated in the Materials & Methods that from 528 HNSCC primary tumor samples, 449 and 443 samples were analysed for differential abundance of bacteria and fungi, respectively. However, the total number of samples in cohort no.2 ("99 HPV+ cancer samples and 427 HPV- cancer samples") do not match 528, 449 nor 443.

2) the numbers of microbes mentioned in the text (line 156 to 163, 170 to 171, 180 to 183, line 224 to 226) do not match the ones shown in Figure 3, Figure 4 and Figure 5.

3) The labels on the axis of each box plot in Figure 3 and Figure 4 are too small for readers to deduce the levels of abundance and the extent of differences among groups.

4) Please clarify in the Materials and Methods regarding group comparison made to generate the results of "Dysregulated in HPV+ HNPCC non-smokers" shown in Figure 6.

Author Response

Please see the point-by-point response below.

Reviewer 2 Report

The paper ijms-1938829 by Chakladar et al reports findings from analyses of the well-known TCGA data focusing on extracting raw whole transcriptome RNA-sequencing data for analyzing microbial contamination or presence of microbial RNA in the samples. They report that the intratumor microbiome is characterized by HPV, smoking and alcohol etiologies in head and neck squamous cell carcinoma. However, there are a number of inaccuracies as the authors report classification of HNSCC into cohorts (HPV+ tumor samples, HPV- tumor samples, HPV- smokers, HPV- nonsmokers, heavy drinkers, and non-drinkers) probably based on categories provided by TCGA but ignoring substantial overlap and uncertainty in these classifications. Localization of the primary HNSCC either in larynx, hypopharynx, oropharynx or the oral cavity is relevant for metastasis, treatment and outcome. Age at diagnosis, sex, etc. are ignored by the authors but may have impact on distibution of risk factor profiles. Treatment modalities etc. pay attention to characteristics of the individual patient. Survival of patients should be analyzed as time-dependent data by Kaplan-Meier curves and Cox proportional hazard models, and not by using the Kruskal-Wallis test for survival status reported for variable follow-up time. Looking at the multitude of data derived from RNA-seq and the multitude of variables analyzed, corrections for multiple testing are required, and a p value <0.05 as threshold for significant findings is inappropriate.

Some of the Figures are not readable. The paper is not ready for publication and requires major revisions and reanalysis.

Author Response

(The authors gave the same response as above.)
